

# Efficient skyline query processing with user-specified conditional preference

Senfu Ke[1], Xiaodong Fu[1,2] and Jie Li[1]

[1] Faculty of Information Engineering and Automation, Kunming University of Science and Technology, Kunming, China
[2] Yunnan Key Laboratory of Computer Technology Application, Kunming University of Science and Technology, Kunming, China

## ABSTRACT

In the realm of multi-attribute decision-making, the utilization of skyline queries has gained increasing popularity for assisting users in identifying objects with optimal attribute combinations. With the growing demand for personalization, integrating user's preferences into skyline queries has emerged as an intriguing and promising research direction. However, the diverse expressions of preferences pose challenges to existing personalized skyline queries. Current methods assume that user preferences are too simplistic and do not represent the interdependencies between attributes. This poses a challenge to the existing skyline methods in effectively managing complex user preferences and dependencies. In this article, we propose an innovative and efficient method for skyline query processing, leveraging conditional preference networks (CP-Nets) to integrate specific user's conditional preferences into the query process, termed as CP-Skyline. Firstly, we introduce a user-defined conditional preference model based on CP-Nets. By integrating user's conditional preference information, we prune the candidate dataset, effectively compressing the query space. Secondly, we define a new dominance relation for CP-Skyline computation. Finally, extensive experiments were conducted on both synthetic and real-world datasets to assess the performance and effectiveness of the proposed methods. The experimental results unequivocally demonstrate a significant enhancement in skyline quality, and it presents a practical and potent solution for personalized decision support.

## INTRODUCTION

Skyline queries have gained significant attention in database management systems due to their ability to identify optimal solutions for multi-criteria decision-making problems. These queries return a set of non-dominated points, called the skyline, that represents the best trade-offs among multiple attributes (*Borzsony, Kossmann & Stocker, 2001*; *Han, Jianzhong Li & Wang, 2013*). In optimization problems with multiple conflicting objectives, finding the skyline solutions provides a set of trade-offs solutions that are not dominated by any other feasible solutions. This approach facilitates decision-making in identifying optimal solutions. Traditionally, skyline query is non-personalized, which does

Corresponding author
Xiaodong Fu,
xiaodong_fu@hotmail.com

not consider the user's preferences and only looks for all non-dominated points, so its results are more objective and comprehensive. However, the conventional skyline query method typically overlooks user-specific preferences, and the user's expected results may vary according to individual needs and expectations.

Consider a scenario where the user seeks the optimal hotel in Table 1 without taking into account the hotel's location. The user's preference for selecting hotels is in the price range between 70 and 120, and the score range is between 5 and 10. Using a skyline query with no preference we need to find non-dominated points in all hotels, the result $R_1 = \{h_1, h_2, h_4, h_{10}\}$, because they do not dominate each other, that is, they are not weaker than each other in both dimensions of price and score. The user identifies the most preferred hotel in $R_1$. If we add user's preference into the skyline query, it is necessary to find only the non-dominated points within the specified preference range (*i.e.*, $h_2, h_3, h_5, h_4, h_6, h_7$) given by the user, and the final query result $R_2 = \{h_2, h_4\}$.

It is evident that if a user's preferences are not considered when generating skyline results, it may include points that are irrelevant or suboptimal to user's needs. In such instances, the system is compelled to search for skyline points within a larger candidate set, requiring users to exert additional effort to filter out uninteresting points from the results. Moreover, as the dataset size increases, the resources required also significantly escalate.

To address this issue, personalized skyline queries have been introduced to filter out irrelevant points for users by integrating their preferences into skyline queries. However, the diverse ways to express user's preferences pose a challenge for preference-based skyline queries. Current personalized skyline query methods often assume linearity and monotonicity in user's preferences (*Ding et al., 2018*). In practice, user preferences may exhibit inter-dependencies between parameters (*Wang et al., 2012*). For example, when choosing a hotel, a user might express preferences such as, "I am looking for a hotel priced below 150. If the price is below 100, a score above 5 is acceptable. However, if the price exceeds 100 but remains under 150, the score should be above 8." This preference style is classified as conditional, where the user's preference for each attribute is contingent. The preference for the price below 150 is prioritized, and the score preference is influenced by the price. Consequently, existing skyline methods might encounter challenges in effectively managing user's intricate preferences and dependencies.

The purpose of this article is to provide a personalized conditional preference skyline calculation scheme, using conditional preference networks (CP-Nets) (*Boutilier et al., 2004*) to handle user preferences and calculate skyline on this basis. CP-Net, as a preference-based conditional probability network, stand out in portraying intricate preference relationships across multiple attributes, including dependencies. *Wang et al. (2015, 2012, 2008)* propose usage condition preferences in the field of personalized service selection, and WCP-nets was proposed for representing and reasoning about conditional preferences in service composition, enhancing the expressive power of user preference. The significant advantage of integrating CP-Nets into skyline queries lies in their ability to manage complex preference relationships, effectively serving as a constrained skyline. We introduce an innovative skyline query methodology based on conditional preferences, integrating user-specific conditional preferences to enhance efficiency and accuracy. By

| Table 1 Relevant hotels. | | | |
|---|---|---|---|
| Hotels | Price | Score | District |
| $h_1$ | 76 | 2 | Dongcheng |
| $h_2$ | 89 | 7 | Haidian |
| $h_3$ | 92 | 5 | Chaoyang |
| $h_4$ | 99 | 8 | Chaoyang |
| $h_5$ | 102 | 6 | Chaoyang |
| $h_6$ | 112 | 3 | Dongcheng |
| $h_7$ | 115 | 5 | Dongcheng |
| $h_8$ | 127 | 2 | Haidian |
| $h_9$ | 136 | 5 | Haidian |
| $h_{10}$ | 157 | 9 | Dongcheng |

adapting to the dependencies in user preferences for tuple's attributes, the proposed approach ensures a more effective and user-centric decision-making process, ultimately elevating the overall user's experience, which can be applied to service selection (*Wang et al., 2015*) or recommendations such as Netflix and Amazon (*Bartolini, Zhang & Papadias, 2011*). By pruning tuples that do not meet user preferences, this approach effectively minimizes computational time and enhances overall processing efficiency, allowing users to quickly find the best options that suit their needs. Moreover, the implementation code is publicly available on GitHub (https://github.com/aSmalltooth/CP-Skyline-Computation/).

The structure of this article is systematically outlined as follows: "Related Work" provides a review of related work on skyline queries and conditional preferences. "Problem Description" presents the formalization of the problem and the definition of CP-Skyline. "CP-Skyline Computation" describes the proposed algorithm for processing skyline queries with conditional preferences and its complexity analysis. "Experimental Evaluation" discusses the experimental evaluation of our approach. Finally, "Conclusion" concludes the article and suggests future research directions.

# RELATED WORK

In this section, we review the mainstream skyline algorithms, especially the personalized skyline algorithms, and discuss the application of skyline technology in some new scenarios.

## Skyline queries

Skyline queries have been extensively researched within the database community as an effective technique for multi-criteria decision-making. The concept of skyline queries was first introduced by *Borzsony, Kossmann & Stocker (2001)*. The basic idea is to find the skyline, which consists of those tuples that are not dominated by any other tuple on all attributes, in the multi-attribute data space. They also proposed the Block nested Loop (BNL) algorithm for efficient skyline computation. Since then, various algorithms have

been proposed to improve the efficiency of skyline query processing, such as the Sort-Filter-Skyline (SFS) algorithm (*Han, Jianzhong Li & Wang, 2013*), the Divide and Conquer (DC) algorithm (*Kossmann, Ramsak & Rost, 2002*), and the Branch and Bound Skyline (BBS) algorithm (*Papadias et al., 2005*).

Over the years, the research on skyline queries includes single skyline query processing algorithms, which assume no user preferences. *Borzsony, Kossmann & Stocker (2001)*, *Papadias et al. (2005)*, and multiple skyline query processing algorithms, which cater to diverse user preferences (*Yuan et al., 2013*). Additionally, skyline query processing has been explored in various application environments such as network information systems, P2P networks (*Stefanini, Palo & Berger, 2024*; *Wang et al., 2007*), and mobile road networks (*Li et al., 2021*; *Cai et al., 2021*). Recent advancements include distributed skyline query processing for big data (*Kuo et al., 2022*; *Bai et al., 2022*), dynamic skyline query processing for changing data (*Zhang et al., 2022*; *Wang et al., 2023*), probabilistic skyline query processing for uncertain data (*Lai et al., 2020*; *Kuo et al., 2022*), and personalized skyline query processing considering user preferences (*Benouaret et al., 2021*; *Bartolini, Zhang & Papadias, 2011*).

## Personalized skyline queries

Due to the fact that traditional skyline query methods do not sufficiently take into account user's personalized preferences, this may result in too many query results failing to meet user's needs. One effective strategy involves minimizing the volume of query results through imerging the user's preferences.

Therefore, many work have made great efforts in the research of skyline queries incorporating user-defined preferences. *Liu et al. (2018)*, *Yuan et al. (2013)* have defined a new type of skyline operation, user-centric skyline computation, which allows users to select skyline points from a dataset based on their preferences for each dimension. *Zhang et al. (2023)* have proposed a user-defined skyline query, which adds a personalized definition of the user's requirements for attribute values to the method and adding user-defined constraints to the subspace skyline query. *Benouaret et al. (2021)* requires users to propose the specific range of data to be searched to represent their preferences, using the value of user's preference/relevant services (matching degree) to represent the degree to which an attribute meets a user's preferences, and then using this matching degree to replace the specific attribute in the skyline calculation. Therefore, incorporating user-defined preferences into skyline queries is an interesting research hotspot that has attracted the attention of many people.

In recent years, the exploration of personalized skyline queries has seen expansion to encompass emerging challenges like uncertainty and incomplete environments. In a centralized computing environment, the definition of uncertain dimensions based on user preferences was introduced by *Saad et al. (2021)*, who employed the skyline query on uncertain dimensions (SkyQUD) algorithm. This algorithm was utilized for partitioning data based on the characteristics of each dataset and conducting skyline probability dominance tests, while the handling of large-scale skyline queries was facilitated through the application of thresholds. In parallel computing environments, the skyline preference

query (SPQ) algorithm for incomplete dataset preference skyline queries was introduced by *Wang et al. (2017)*. Their innovative strategy involved the segregation and classification of massive incomplete datasets based on the importance of dimensions, resulting in a significant enhancement of the efficiency of skyline queries performed on such datasets. Considering user's preferences, data objects were partitioned into different grids using Voronoi diagrams by *Tai, Wang & Chen (2021)*. They conducted parallel computations on object combinations to acquire dynamic skyline results. Meanwhile, Top-k skyline preference queries under MapReduce were proposed by *Zheng et al. (2021)*. Datasets were segmented into regions based on user-preferred dimensions, local skyline sets were computed in parallel, and relaxed dominance comparisons between regions were used to derive the global skyline set.

In essence, existing skyline query research mainly focuses on processing techniques and applications. Preference-driven skyline queries have gained widespread attention for their potential in reducing costs and improving user's experiences. However, current methods often oversimplify user's preferences, neglecting intricacies like dependencies between attributes. This article addresses these issues using CP-Nets, enhancing both query efficiency and result quality. This method supports reasoning based on local preferences and can effectively handle the problem of query time increasing with increasing dimensions.

## PROBLEM DESCRIPTION

In this section, we present the fundamental concepts utilized in this article and provide a formal definition of CP-Skyline. Furthermore, we illustrate an example of employing CP-Nets to model user's preferences. The calculation process of finding CP-Skyline tuples will be subsequently elaborated upon based on this example. Additionally, a comprehensive summary of the frequently used symbols and their corresponding descriptions is presented in Table 2.

### Representing user's preference with CP-Nets

To incorporate user conditional preferences into skyline queries, several fundamental issues need addressing: representing user preferences, the methods or tools used to model these preferences, and reasoning about preference statements (*Mouhoub & Ahmed, 2018*). CP-Nets offer a natural and clear way to represent conditional preferences. Structurally, CP-Nets form a directed graph where vertices represent tuple attributes and directed edges indicate dependencies between attributes. Each vertex's conditional preference table shows user preferences for different attribute values. CP-Nets also express dominance relationships between outcomes through sparse attribute dependencies (*e.g.*, a *Price* less than 150 with a *Score* greater than 7 is preferred over a *Price* of 100 with a *Score* greater than 5).

In CP-Skyline query process, we allow users to provide preferences that are more detailed than the constrained subspace (*Dellis et al., 2006*), *i.e.*, incorporating dependency relationships across different dimensions. CP-net is used to represent the user's conditional preference, and the user's preference for data is derived from the semantics

**Table 2 Notation.**

| Symbol | Description |
|---|---|
| $T, t_i$ | Set of tuples, a specific tuple |
| $A, a_p$ | Set of attributes, a specific attribute |
| $H, h_x$ | Set of hierarchies, a specific hierarchy |
| $\Omega, o_i$ | The outcome space, a specific outcome |
| $m$ | The number of attributes |
| $m'$ | The number of user's specified attributes |
| $CPT(a_p)$ | Conditional preference table of $a_p$ |
| $\succ_u^i$ | Preference order of $CPT(a_p)$ |
| $t_i[a_p]$ | The value of $t_i$ on attribute $a_p$ |
| $t_i \succ t_j$ | Tuple $t_i$ strictly dominates tuple $t_j$ |
| $t_i \succ^H t_j$ | Tuple $t_i$ hierarchy dominates tuple $t_j$ |
| $t_i \succ^{CP} t_j$ | Tuple $t_i$ CP-dominates tuple $t_j$ |
| $CS$ | Candidate set |
| $CPS$ | CP-Skyline points |

and properties of CP-net. Therefore, we start by introducing some notions from decision theory and the CP-nets formalism for preference representation. In addition, CP-Nets can be learned from the preference database, which is not involved in this article (*Alanazi, Mouhoub & Zilles, 2020*). So, we start by introducing some notions from decision theory and the CP-nets formalism for preference representation.

**Definition 1** (CP-Net; *Boutilier et al., 2004*). *A CP-Net over attributes $A = \{a_1, \ldots, a_m\}$ is a directed graph G over $a_1, \ldots, a_m$ in which each node is annotated with conditional preference tables $CPT(a_p)$. Each conditional preference table associates a preference order $\succ_u^i$ with each instantiation $u$ of $a_p$'s parents $Par(a_p) = U$. The conditional preference rule (CPR) on variable $a_p$ is an expression: $u : [a_p] \succ [a_p']$, where $u$ is the assignment of $U$, $[a_p]$, $[a_p']$ is the assignment of the attribute $a_p$. Such a rule means "assuming $u$ holds, all other variables being equal, $[a_p]$ is better than $[a_p']$". The conditional preference table $CPT(a_p)$ on attribute $a_p$ is the set of CPRs on $a_p$.*

Given a simple example, the preference for hotel selection, to illustrate how to use CP-Nets to represent the user's conditional preferences:

**Example 1** (CP-Net). *Consider a traveler want to find hotel, his complete preference for the trip is provided shows as Fig. 1, and the details of his preferences are summarized as follows:*

*(1) Assume that there are three options for where the hotels located: it can be located in Dongcheng District, Chaoyang District or Haiding District. The user's preference over the options is unconditional: Dongcheng District $\succ$ Chaoyang District $\succ$ Haiding District, where "$\succ$" denotes one is more preferred than the other.*
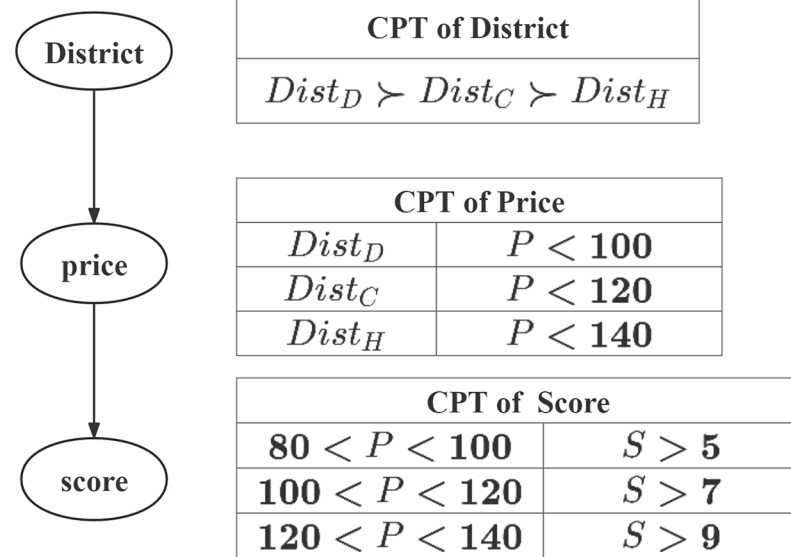

**Figure 1** CP-Net for Example 1.

*(2) Assume the user's preference over the Price is conditional: Dongcheng District: 80 < Price < 100; Chaoyang District: Price < 120; Haiding District: Price < 140. And the lower the Price, the better.*

*(3) Assume the user's preference over the Score is conditional: 80 < Price < 100: Score5; 100 < Price < 120: Score7; 120 < Price < 140: Score9. And the higher the Score, the better.*

Now we apply the above notions and definitions to create a CP-Net for representing user preferences in our Travel example, and the CP-Nets and *CPT* shown in Fig. 1 according to the user's preferences. The arrows in Fig. 1 mean the relations among the attributes. For example, *Price* depends on the values of *District*, *Score* depends on the values of *Price*.

Next, we formally define CP-Skyline and the dominance relations involved in the calculation process.

## Formal definition of CP-skyline

**Definition 2** *(Strictly Dominance). Given a dataset T, we say that a tuple $t_i$ dominates another tuple $t_j$, denoted as $t_i \succ t_j$ if and only if $t_i$ has better performance than $t_j$ on all specified preference attributes, and strictly better performance on at least one attributes. i.e., $t_i \succ t_j \Leftrightarrow \forall a_p \in A : t_i[a_p] \succeq t_j[a_p] \land \exists a_q \in A : t_i[a_q] \succ t_i[a_q]$.*

For example, hotels in Table 1 meet $h_4 \succ h_5$ if and only if $h_4$ is superior to $h_5$ in at least one attribute.

**Definition 3** *(Hierarchy Dominance). Given the user's preference and divides it into different preference levels (e.g., the most interested, and second interested, etc.), which termed as the hierarchy. Given a dataset T, we say that a tuple $t_i$ hierarchy dominates another tuple $t_j$, denoted as $t_i \succ^H t_j$, if and only if $t_i$ is in hierarchy $h_x$, $t_j$ is in another hierarchy $h_y$, and $h_x$ have a higher preference level than $h_y$ for user. i.e., $t_i \succ^H t_j \Leftrightarrow \forall t_i \in h_x, \forall t_j \in h_y : h_x \succ h_y$.*

For example, hotels in Table 1 meet $h_5 \in h_x \succ^H h_4 \in h_y$ if and only if the preference level of $h_5$ is higher than the preference level of $h_4$. Because $h_4$ is better than $h_5$ in both price and score, if $h_5$ is better than $h_4$: for the user, the preference level of $h_5$ is higher than $h_4$, for example, the user's preference *price > 100* is better than other prices.

**Definition 4** *(CP-Dominance). Given a m-dimensional dataset T and a user's CP-Net, we say a tuple $t_i$ CP-Dominance another tuple $t_j$, denoted as $t_i \succ^{CP} t_j$, if and only if $t_i$ has better performance than $t_j$ on all specified preference attributes, and strictly better performance on at least one attributes, at the same time, $t_i$'s preference level $h_x$ is not lower than $t_j$'s preference level $h_y$, and the tuples with high preference level unconditionally dominate the tuples with low preference level, i.e.,*

$$t_i \succ^{CP} t_j \Leftrightarrow \forall\, t_i, t_j \in h_x : t_i \succ t_j \vee \ \forall t_i \in h_x, \forall t_j \in h_y : h_x \succ h_y.$$

For example, hotels in Table 1 meet $h_4 \succ^{CP} h_5$ if and only if $h_4$ and $h_5$ are in the same preference level, at this time, for $h_4$ and $h_5$, users have no obvious preference, so they only need to judge based on attribute values ($h_4 \succ h_5$). And $h_5 \succ^{CP} h_4$ if and only if the preference level of $h_5$ is higher than the preference level of $h_4$, at this time, for $h_4$ and $h_5$, the importance of user preference is greater than the quality of attribute values, so $h_5$ with a high preference level is better than $h_4$. ($h_5 \succ^H h_4$).

In other words, CP-Dominance satisfies both Strictly Dominance and Hierarchy Dominance.

**Definition 5** *(CP-Skyline). Given a dataset T and a user's CP-Net, the CP-Skyline comprises the set of tuples are not CP-Dominated by other tuples. i.e., $CPS = \{t_i \in T | t_j \in T : t_i \succ^{CP} t_j\}$.*

We now provide the formal definition of the problem of using CP-Nets to represent user's conditional preferences.

**Definition 6** *(Problem Statement). Given user's personalized Conditional Preference (expressed as CP-Nets) and a dataset T on a set of attributes A defined by a set of m attributes $\{a_1, \ldots, a_m\}$, a tuple object $t_i \in T$ is represented as an m-dimensional tuple $t_i = \{t_i[a_1], \ldots, t_i[a_m]\}$ where $t_i[a_p]$ is the value on attribute $a_p$. Based on this information, we calculate the CP-Skyline set (CPS) based on user's conditional preference, that is, the set of tuples that are not CP-Dominaned by other tuples.*

## CP-SKYLINE COMPUTATION

Our algorithm unfolds in two distinct stages: the initial phase, denoted as the Pruning stage, aims to refine the candidate set based on CP-Nets semantics, thereby identifying tuples that align with user's preferences. The subsequent stage, computes the CP-Skyline from the pruned tuples, ultimately yielding the final skyline set.

Firstly, in the stage 1, the candidate set is pruned based on the semantics of CP-Nets to derive a set of tuples that satisfies all user's preferences. The preference induced graph (*Li, Vo & Kowalczyk, 2011*) from the user's CP-Nets enables us to ascertain the relative importance of various outcomes to the user. Subsequently, a traversal of all tuples is conducted for classification. At this stage, each tuple within the candidate set is allocated to a specific category. These categories are then sorted according to the induced graph derived from the user-provided CP-Nets, referred to as a hierarchy, representing the user's

preference levels for different outcomes. In the stage 2, the CP-Skyline is computed based on the definition of CP-dominance.

## Pruning strategy

The pruning strategy in stage 1 involves two processes. Firstly, it prunes the candidate set using CP-Nets semantics, filtering out tuples that do not meet preferences and assigning satisfying tuples to different preference levels *via* a classification tree. Secondly, it uses the induced graph from CP-Nets to determine the significance of each hierarchy to the user, allowing us to sort the hierarchies and select tuples only from the most satisfactory level as the candidate set, thus achieving pruning effects.

### *Construction of classification tree*

A classification tree segmentally categorizes tuples based on a sequence of conditional checks. Its fundamental concept involves partitioning the feature space into rectangles. Starting from the root node, the tuple is progressively separated through a series of decisions. When the division is complete, and the tuple cannot be further separated, it forms a leaf node.

**Definition 7** (*Classification tree*). *Given a CP-Net $G$ over $A = \{a_1, a_2, \ldots, a_m\}$, a classification tree $C = (V, E_1)$ structured by $G$. Among $C$, the node set $V$ contains all the attribute nodes corresponding to the tuple attributes. Each attribute node $a_p \in A$ represents a specific attributes of the tuple. The edge set $E_1$ describes the causal relationship and conditional dependence between the various attributes. For each attribute node, the set of its parent nodes is denoted by $Par(a_p) = U$, and for all possible parent node value combinations $u \in U^{|Par(a_p)|}$. Related to it is a conditional preference table $CPT(a_p)$, the table specifies that when the parent node value of the attribute $a_p$ is $u$, the user pairs the attribute $a_p \succ_u^i$ represents preference ordering of attributes possible values.*

Constructed upon this foundation, the classification tree exhibits the following characteristics:

*Root node.* The root node, *District* is chosen due to its independence from other attributes, eliminating the need for a *CPT*.

*Internal nodes.* Each root node value corresponds to an internal node within a sub-tree, representing geographical branches. For attributes like *Price* and *Score*, internal nodes are determined by the associated *CPT*, reflecting user preferences under specific conditions.

*Leaf nodes.* Leaf nodes represent unique attribute configurations and detailed conditional preferences, indicating that a candidate tuple aligns with the user's overall preference profile.

*Edges.* Each directed edge from node $v_i$ to node $v_j$, denoted as $(v_i, v_j) \in E_1$, shows that the optimal choice for the next related attribute can be inferred from the *CPT* of node $v_j$. These edges demonstrate the dependency structure in the decision process.

Upon constructing the classification tree, each $t_i \in S$ is traversed starting from the root node. At each internal node, the attribute value of tuple is checked against the current criterion. If it meets the condition, traversal continues along the relevant branch; otherwise, "No Valid Category" is returned (represented by $F$ in classification tree), or the

process halts. Tuples that satisfy the user's conditional preferences are classified ($o_i \in \Omega$), while those failing the CP-Nets criteria are pruned. This completes the first pruning step, resulting in $CS = \Omega$.

Using the CP-Net from Example 1 and tuples from Table 1, a classification tree is constructed as shown in Fig. 2. Tuples $h_1, h_{10}$ are excluded due to their *Price* being outside $[80, 140]$, and $h_1, h_6, h_8$ are excluded because their Scores are not greater than 5. Consequently, each tuple is assigned to a classification that meets the user's conditional preferences.

According to the user's CP-Nets, the most important attribute is the administrative *District*, followed by *Price*, with *Score* being the least important. The classification tree removes tuples not meeting the preferences (*e.g.*, $t_5$ does not satisfy: if $100 < Price < 120$, then *Score* 7). Tuples satisfying the preferences are assigned to specific categories ($o_i$), such as $t_2 \in o_4$ and $t_3, t_4 \in o_2$. At this stage, non-compliant tuples are removed from the candidate set, resulting in $CS = \Omega = \{o_1, o_2, o_3, o_4, o_5, o_6\}$.

### Construction of induced graph

Presently, through the utilization of a classification tree, we can filtered out tuple that does not align with user's preferences. Our current focus lies in sorting $o_i$ based on varying degrees of user's satisfaction. As a precursor to pruning, we introduce the concept of the induced graph, outlining the procedure for attaining optimal results using CP-Nets.

**Definition 8** (*Induced graph Li, Vo & Kowalczyk, 2011*). *Given a acyclic CP-Net G, the Induced preference graph donated by $I = (\Omega, E_2)$, is defined by following edges between outcomes, where $E_2$ is the user's preference for the two outcomes. For each attribute $a_p$, $o, o' \in \Omega$ that differ only on the value of $t[a_p]$, let there be a directed edge $o' \rightarrow o$ if a user prefer $o$ to $o'$; there be a directed edge $o \rightarrow o'$ if a user prefer $o'$ to $o$. If $o$ and $o'$ are incomparable, $I$ does not contain any edge between $o$ and $o'$.*

Based on the current preference-induced graph, we can rank the importance of different outcome $o_i \in \Omega$ to the user. Selecting the outcomes with the highest preference level $o_i$ as $CS$ allows us to perform the second step of pruning.

According to Definition 7, the induced graph is constructed as shown in Fig. 3. The node set, $\Omega$, represents different outcomes. The edge set, derived from the user's CP-Nets, includes directed edges $<o_i, o_j>$, where $o_i$ and $o_j$ differ by only one attribute value. For example, the preference for $o_1(Dist_D \wedge P < 100 \wedge S > 5)$ over $o_2(Dist_C \wedge P < 100 \wedge S > 5)$, indicated by the directed arc, reflects the semantics of $CPT$ (*District*).

Conducting a topological sort on the preference-induced graph can determine the ranking of outcomes: $o_1 \succ o_2 \succ o_3 \succ o_4 \succ o_5 \succ o_6$. The top-level node ($o_6$) represents the least preferred outcome, while the bottom-level node ($o_1$) represents the most preferred. We select $o_1$, the outcome to the user is most interested in, as $CS$. If $o_1$ is empty, the next best $o_2$ is chosen as $CS$. After determining $CS$, we proceed with the CP-Skyline calculation.

### Pruning

Pruning based on the above two process are presented in Algorithm 1 (Line 4). The algorithm aims to systematically derive a candidate set ($CS$) from an acyclic CP-Net $G$ and

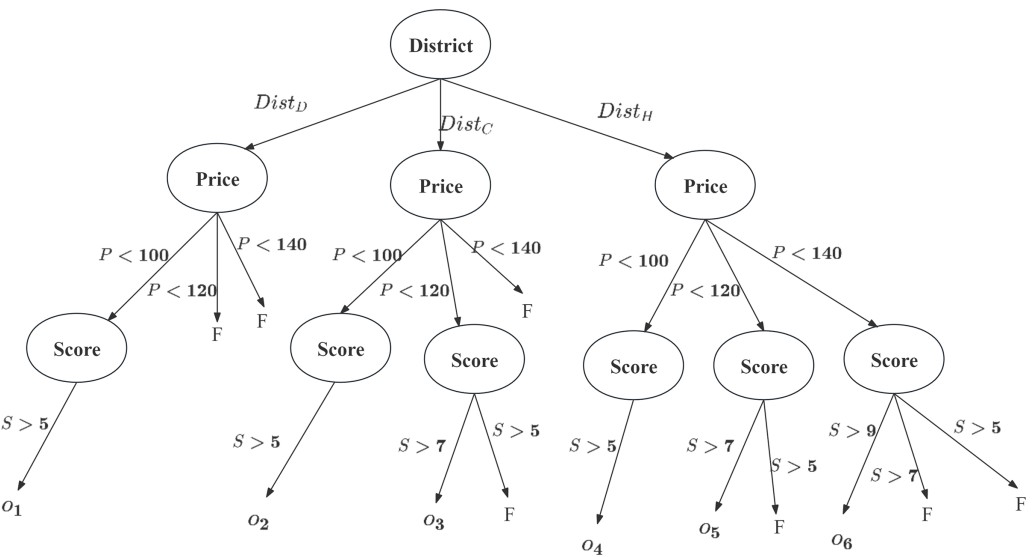

**Figure 2  Classification tree for this case.**

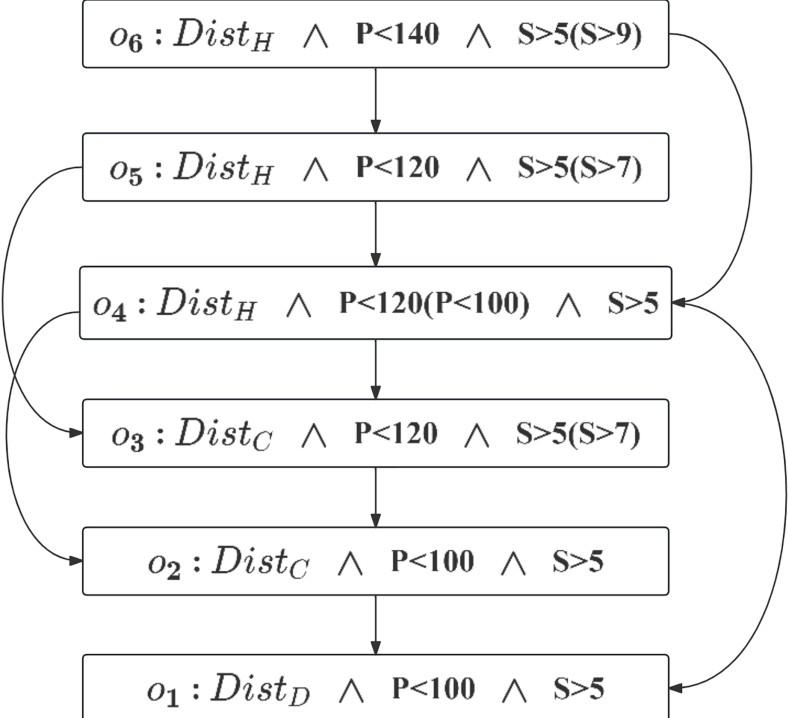

**Figure 3  Induced preference graph.**

a set of tuples $T$. The algorithm commences by initializing $CS$ and Hierarchies as empty sets. Subsequently, it constructs a classification tree $C$ and a preference induced graph $I$ based on the characteristics of the given acyclic CP-Net (line 5). The algorithm then enters a loop that iterates through tuples $t_i$ in $T$ which align with the branches of the classification tree $C$. For each tuples, it categorizes it into various outcomes ($o_i \in \Omega$). The identified $\Omega$

**Algorithm 1  CP-Skyline computation Algorithm.**

**Input:** Acyclic CP-Net $G$, dataset $T$

**Output:** set of CP-Skyline points ($CPS$)

1: Initialize *Hierarchies* $\leftarrow \varnothing$

2: Initialize Candidate dataset ($CS$) $\leftarrow \varnothing$

3: Initialize CP-Skyline points ($CPS$) $\leftarrow \varnothing$

4: **Stage 1 (Pruning):**

5: Generate Classification tree $C$ and preference induced graph $I$ by $G$

6: **while** $t_i \in T$ **do**

7:     Classify $t_i$ to different outcomes ($o_i \in \Omega$) by $C$

8: **end while**

9: *Hierarchies* $= \Omega$

10: Sort *Hierarchies* by topological reverse sort of $I$

11: **for** each *hierarchy* $\in$ *Hierarchies* (starting from the highest hierarchy) **do**

12:     **if** *hierarchy* $\neq \varnothing$ **then**

13:         $CS =$ current *hierarchy*

14:     **else**

15:         $CS =$ next *hierarchy*

16:     **end if**

17: **end for**

18: **Stage 2 (Skyline computation):**

19: **for** each $c_i \in CS$ **do**

20:     *dominated* $\leftarrow$ false

21:     **for** each $c_j \in CPS$ **do**

22:         **if** $c_j$ dominates $c_i$ **then**

23:             *dominated* $\leftarrow$ true

24:             **break**

25:         **end if**

26:     **end for**

27:     **if** not *dominated* **then**

28:         Remove all $c_j$ from $CPS$ such that $c_i$ dominates $c_j$

29:         Add $c_i$ to $CPS$

30:     **end if**

31: **end for**

32: **return** $CPS$

that satisfy the user's conditional preferences are subsequently appended to the Hierarchies set. The hierarchies are further sorted in accordance with a topological reverse sort of the preference-induced graph $I$, since the CP-Nets discussed are acyclic, their derivation is also

acyclic and therefore does not form a cycle (*Boutilier et al., 2004*). For example, Fig. 3 shows the acyclic derived graph obtained through acyclic CP-Nets. The topological sorting result is $o_1 \succ o_2 \succ o_3 \succ o_4 \succ o_5 \succ o_6$, that is, the hierarchies set $H = h_1 \succ h_2 \succ h_3 \succ h_4 \succ h_5 \succ h_6$. According to definition 3, it can be seen that no cycle will be formed. Following this, the algorithm iterates through each hierarchy in *Hierarchies*, commencing with the highest hierarchy. If the hierarchy is not empty, it sets *CS* to the current hierarchy. Otherwise, it advances to the next hierarchy. Finally, the algorithm concludes by returning the resulting candidate set *CS*. This descriptive passage outlines the steps and purpose of the "Pruning" algorithm, emphasizing its operations on acyclic CP-Nets and tuples to generate a refined candidate set in accordance with user's preferences.

## Skyline computation stage

The stage 2 is employed for computing the CP-Skyline set (line 18), it takes *CS* from pruning stage as input and produces the set of CP-Skyline set denoted as *CPS*. The algorithm iterates through each candidate $c_i$ in *CS*. For each $c_i$, it checks whether it is dominated by any existing tuple in *CPS*, the current CP-Skyline set. If $c_i$ is not dominated by any tuple in *CPS*, it removes all the tuples in *CPS* that are dominated by $c_i$ and adds $c_i$ to *CPS*.

The complete process is encompassed by Stage 1 and Stage 2, which collectively ensure the CP-Skyline computation algorithm. This process effectively prunes dominated tuples from the candidate set, ensuring that only non-dominated tuples are retained in the resulting CP-Skyline set.

## Computational complexity

The computational cost of our method is the sum of two stages. In the pruning stage, the Classification tree is used to filter and divide the candidate data requires sorting all attributes and calculating each possible split point, which takes the time complexity of each split $O(n_1 \cdot m')$, in the worst case The depth of the classification tree is $O(\log(n_1))$, so the total time complexity is $O(n_1 \cdot m' \cdot \log(n_1))$ ($n_1$ is the number of tuples in $T$). The time taken to obtain the preference induced graph is $O(m')$. During topological sorting, each vertex is processed only once. Each edge is also processed only once, either to update the in-degree of the vertex or to perform a depth-first search (DFS). Therefore, the total time complexity of topological sorting is $O(\Omega + E_2)$. The second stage is to perform CP-dominance calculation on the candidate data set, which requires domination calculation between pairs of data, and the time complexity is $(n_2^2)$, which takes $(n_2^2)$ times ($n_1 \gg n_2$). The complexity of constructing the exported graph and topological sorting it is constant level, and the query dimension and the size of the candidate set data have the greatest impact on it, so the final time complexity is $O(n_1 \cdot m' \cdot \log(n_1) + n_2^2)$. However, in the worst case (*e.g.*, data imbalance), the depth of classification tree may reach $O(n_1^2)$, so the worst case time complexity is $O(n_1^2 \cdot m' + n_2^2)$. In practical applications, while the theoretical worst-case complexity is high, the actual runtime is often much lower. The pruning stage effectively removes tuples that do not meet user preferences early on, significantly reducing the candidate set size and overall computation time. This efficiency

gain means the method performs well in most practical scenarios, despite the high worst-case complexity.

# EXPERIMENTAL EVALUATION

In this section, we first describe the experiment setup and then we present and discuss the respective results. To verify the effectiveness and performance of the CP-Skyline query method, we design and implement relevant experiments and analyze the results. Prior to conducting the comparative experiments, it was imperative to ensure a consistent evaluation framework across all skyline methods. To achieve this, we employed a uniform approach by instructing each method to utilize a brute-force algorithm for computing result sets.

The experimental environment is Intel® Xeon® Gold 6330, 80G RAM, 64-bit Windows 11 professional operating system, PyCharm 2022 development environment, Python 3.8 development language.

## Datasets

To evaluate the effectiveness of CP-Skyline queries on various datasets, experiments were conducted using both real and synthetic data, the proposed method was evaluated on dataset sizes ranging from 10 to 800 k.

The real datasets include:

QWS (https://qwsdata.github.io/): Measures the quality of 2,507 Web services, collected in 2008 using the Web Service Broker framework, with each entry containing nine QWS measurements. Each row in this dataset represents a web service and its corresponding nine QWS measurements (separated by commas). The first nine elements are QWS metrics that were measured using multiple Web service benchmark tools over a 6-day period. The QWS values represent averages of the measurements collected during that period.

NBA Players NBA Players (https://www.kaggle.com/datasets/justinas/nba-players-data/): Contains demographic and biographical details, as well as performance statistics for NBA players over 20 years, 12,843 entries in total. The data set contains over two decades of data on each player who has been part of an NBA teams' roster. It captures demographic variables such as age, height, weight and place of birth, biographical details like the team played for, draft year and round. In addition, it has basic box score statistics such as games played, average number of points, rebounds, assists, *etc*.

Synthetic dataset: data was randomly generated within the range of the QWS parameters' maximum and minimum values, with a total of 800,000 pieces of data.

Synthetic anti-correlated dataset: Synthetic anti-correlated dataset: data was randomly generated within the range of the QWS parameters' maximum and minimum values, with a total of 800,000 pieces of data. This dataset is created by generating random values for each attribute (*e.g.*, availability, throughput) and subtracting them from fixed values to ensure inverse correlation. The resulting dataset contains multiple anti-correlated attributes.

## Experiments results

In the sets of experiments, a comparison was conducted between CP-Skyline and the traditional skyline alongside several personalized skyline queries across various aspects. Including the following methods: *Subspace Skyline* (2015) (*Yuan et al., 2013*), which allows users to provide attributes of interest and only perform queries in the subspace that the user is interested in, involves the simplest user preference. *User-Defined Skyline* (2023) incorporates personalized requirements for attributes values and User-Defined constraints (*Zhang et al., 2023*), and *Decisive Skyline* (2022) (*Vlachou et al., 2022*), which retrieves a set of points that balance all specified criteria, is a tuple that is better under all attributes in the skyline combination, also means a collection of high-quality skyline points. To highlight the query efficiency of CP-Skyline, we control for the same user preference attributes across several personalized skyline query methods, as the form of user preferences expressed by these methods varies.

When conducting comparative experiments, traditional skyline has no user preferences and is calculated in the entire space. Since several personalized skylines can handle different user preferences, we uniformly set user preferences to the most detailed conditional preference among several preferences.

### *Size of the skyline result sets*

The size of the skyline result set is an important metric for evaluating query performance and efficiency according to the standards of scientific articles (*Benouaret et al., 2021*). A smaller result set typically indicates faster query processing, as the system returns only the most significant or valuable data set. By controlling the size of the skyline result set, it is possible to ensure that the returned result contains the most valuable and crucial information, thereby assisting users in making more accurate decisions and enhancing their overall experience.

We first compare the result sets of several skylines on the real data set QWS (Fig. 4A) and NBA players (Fig. 4B). Because the data set of the real data set is too small, we then analyze the changes of several skyline result sets with the size of the data set by synthesizing the data set (Fig. 5). Finally, we analyze the changes of various skyline result sets with the query attribute by changing the query attribute 2–5 on QWS (Fig. 6A) and NBA players (Fig. 6B).

Figures 4A and 4B depicts the size of result set of different skyline query methods on QWS and NBA players, where in this case, the entire dataset was used as the candidate set. Traditional skyline queries were conducted with a full-dimensional query space, while personalized skyline queries were performed with a five-dimensional query space. Obviously, the result set obtained by the traditional skyline query method has the largest number, while the remaining personalized skyline algorithms have a smaller result set, because they exclude tuples that does not meet the personalized requirements. In Fig. 5, synthetic data was utilized to examine the effect of candidate set size on the result set size with a five-dimensional query space. In all cases, the size of the result set will increase with the size of the dataset. However, the size of the result set of the personalized skyline algorithm is smaller than the size of the result set obtained by the traditional skyline query.

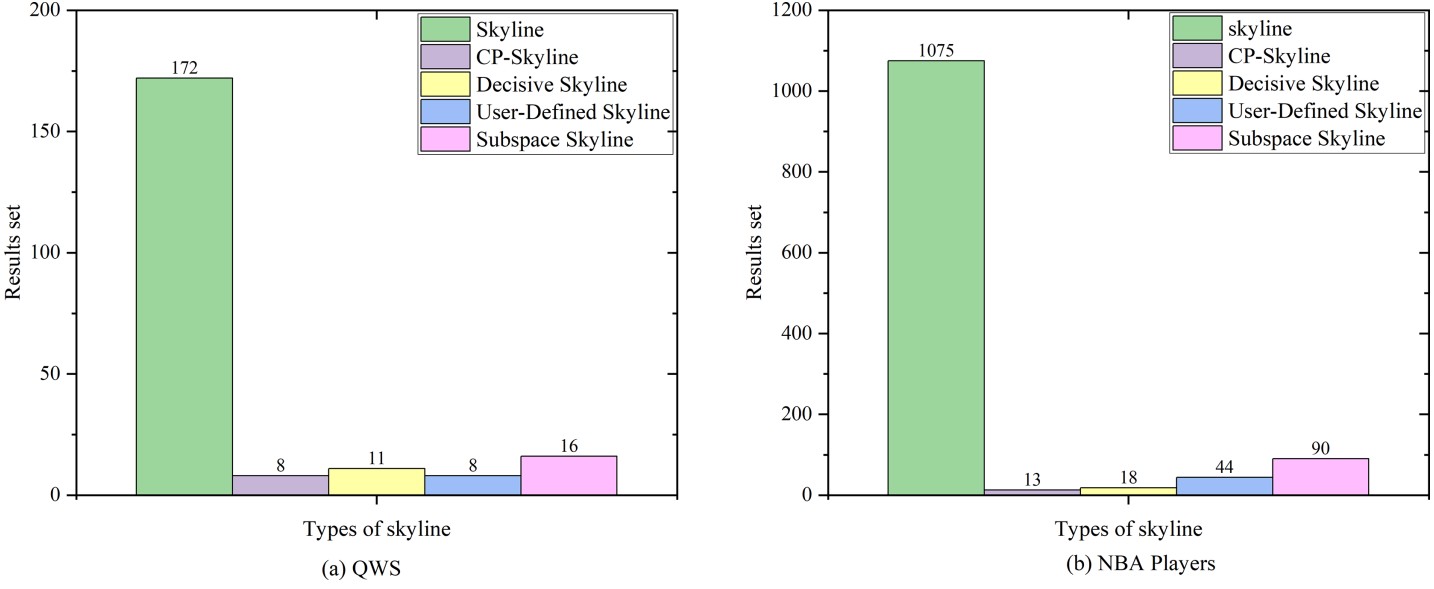

**Figure 4** (A, B) Size of the result set *vs.* number of dimensions on different datasets.

**Figure 5** Size of the result set *vs.* size of dataset (k) on synth dataset.

As illustrated in the figure, CP-Skyline eliminates a lot of tuples that does not meet the user's personalized requirements compared to the traditional skyline query method. Compared with the user-defined skyline, it also has a certain improvement. As the the decisive skyline defined as the subset of skylines that balances all attributes, its size is smaller than CP-Skyline.

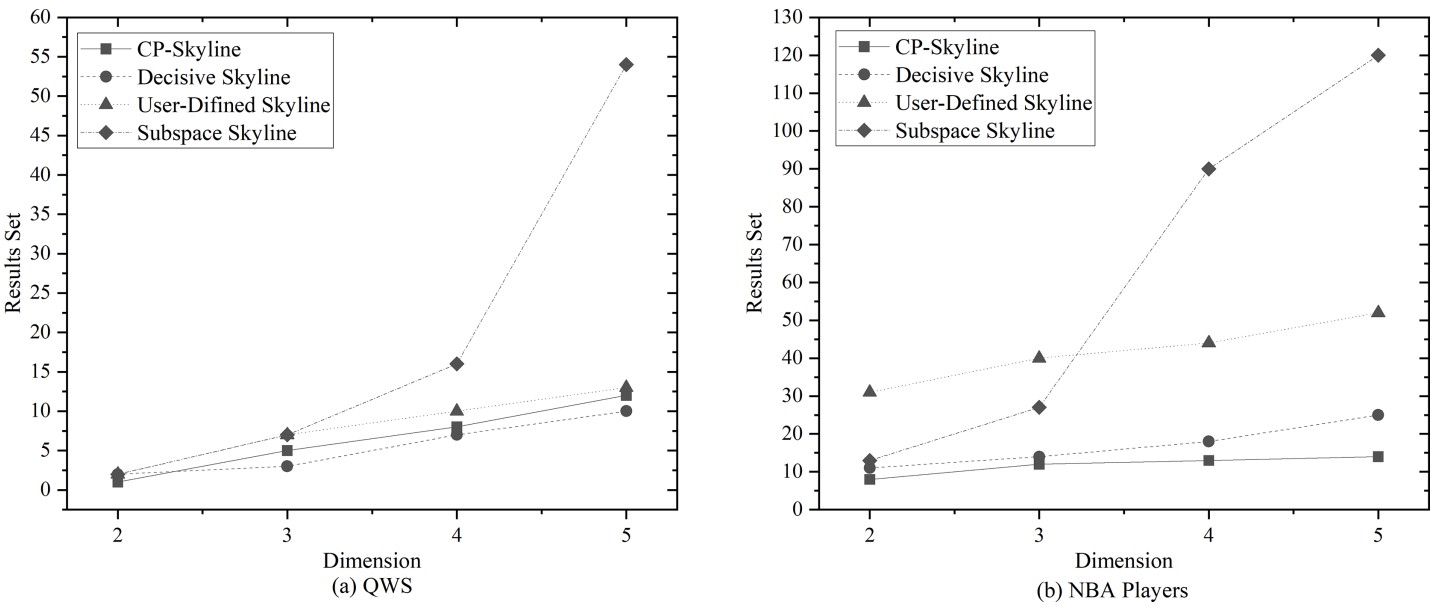

**Figure 6 (A, B) Size of the result set *vs.* number of dimensions on different datasets.**

Next, in Figs. 6A and 6B, the impact of the query dimension (attribute) involved in the skyline on the size of the result set is examined on QWS and NBA players. In all cases, the size of the result set will increase as the number of tuple's attribute increases. Since the skyline is a set that is not dominated by other tuples on at least one dimension, increasing the query dimension means that tuples are more difficult to be dominated. Due to the large volume and dense distribution of data in the NBAPlayers dataset, there may be multiple instances of the same data in low-dimensional cases. The user-defined skyline exclude those instances that can dominate these tuples, resulting in a higher number of undominated tuples, Which leads to the the situation in Fig. 6B. Overall, the personalized skyline query method has a smaller change in the result set. For users, when the query dimension increases, The user's effort to locate the most preferred tuple intensifies, and the more specific the user's preference is, the more tuples that do not meet user preferences are removed through pruning, the less likely these tuples are to become part of the Skyline. As a result, the smaller the result set is, and the less effort is spent.

### Skyline query response time

In the skyline query algorithm, because it deals with large-scale multi-dimensional data sets, the running time of the algorithm will affect the efficiency of the entire query process. In practical applications, the shorter running time means that the algorithm can respond to user's query requests faster and improve the responsiveness and user's experience of the system. And the size of the data set may continue to expand, so the running time of the algorithm will be affected by the size of the data. If the running time of the algorithm increases significantly with the increase of data size, the scalability of the algorithm may be affected and cannot meet the needs of future data size growth.

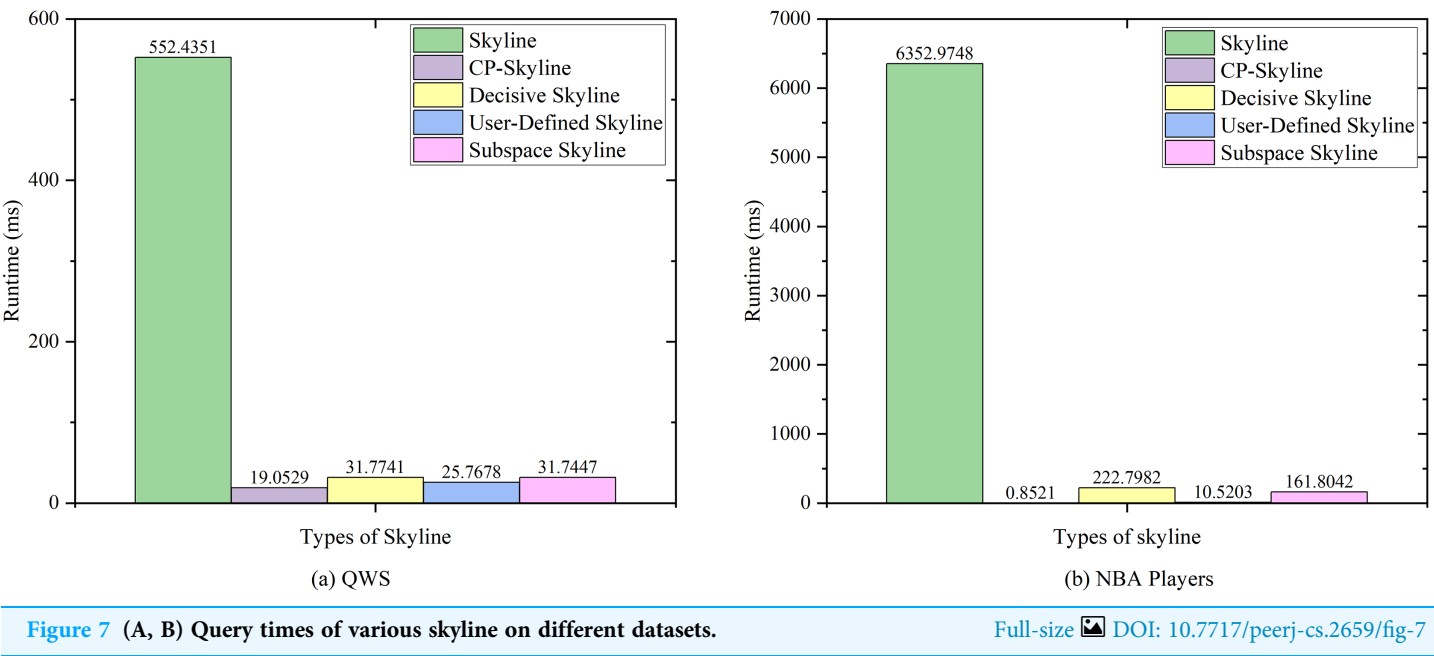

**Figure 7 (A, B) Query times of various skyline on different datasets.**

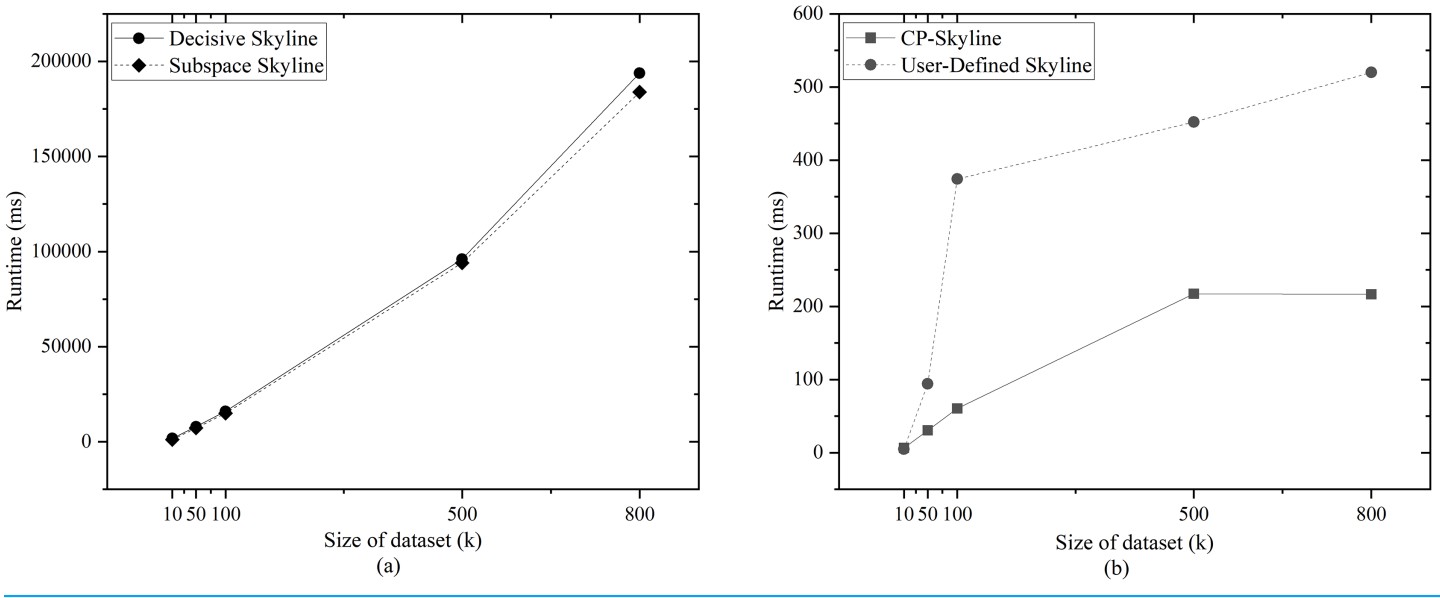

**Figure 8 (A, B) Query time *vs*. size of dataset (k) on synth dataset.**

We first compare the query time of several skylines on the real data set QWS (Fig. 7A) and NBA players (Fig. 7B). Then we analyze the changes of query time of several with the size of the data set by synthesizing the data set (Fig. 8). We analyze the changes of query time of various skyline with the query dimension by changing the query dimension 2–5 on

**Peer**J Computer Science

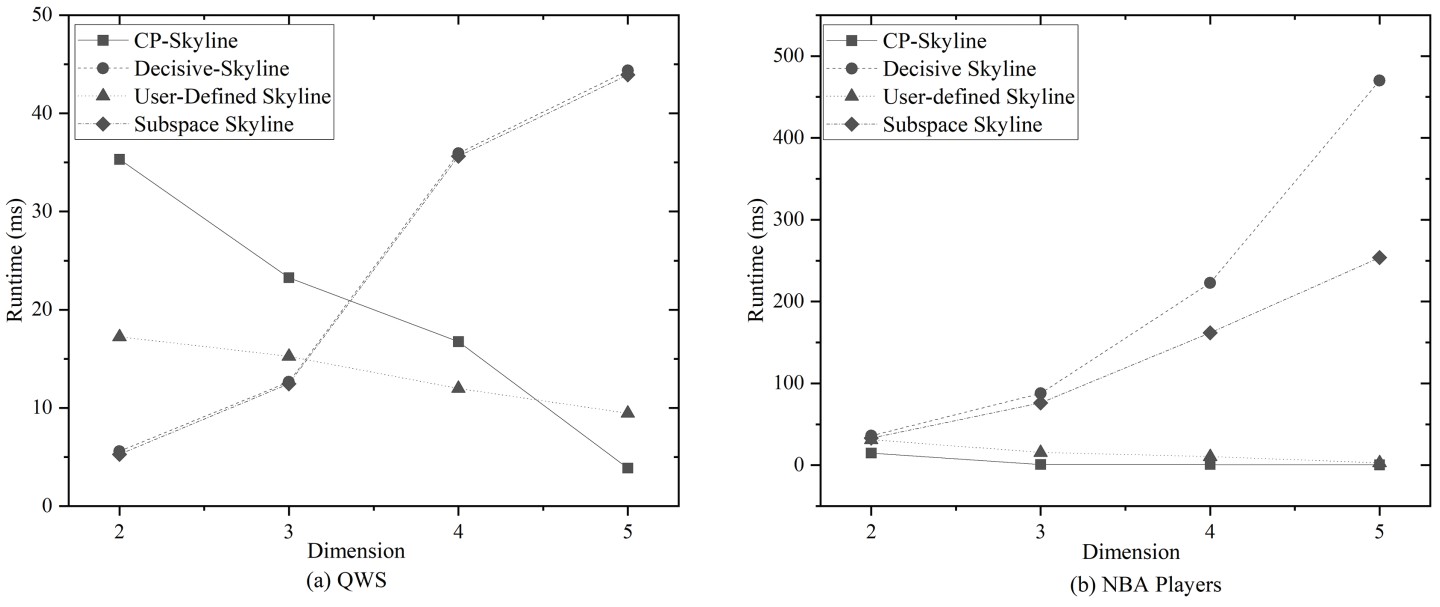

**Figure 9  (A, B) Query time *vs*. number of dimensions on different datasets.**

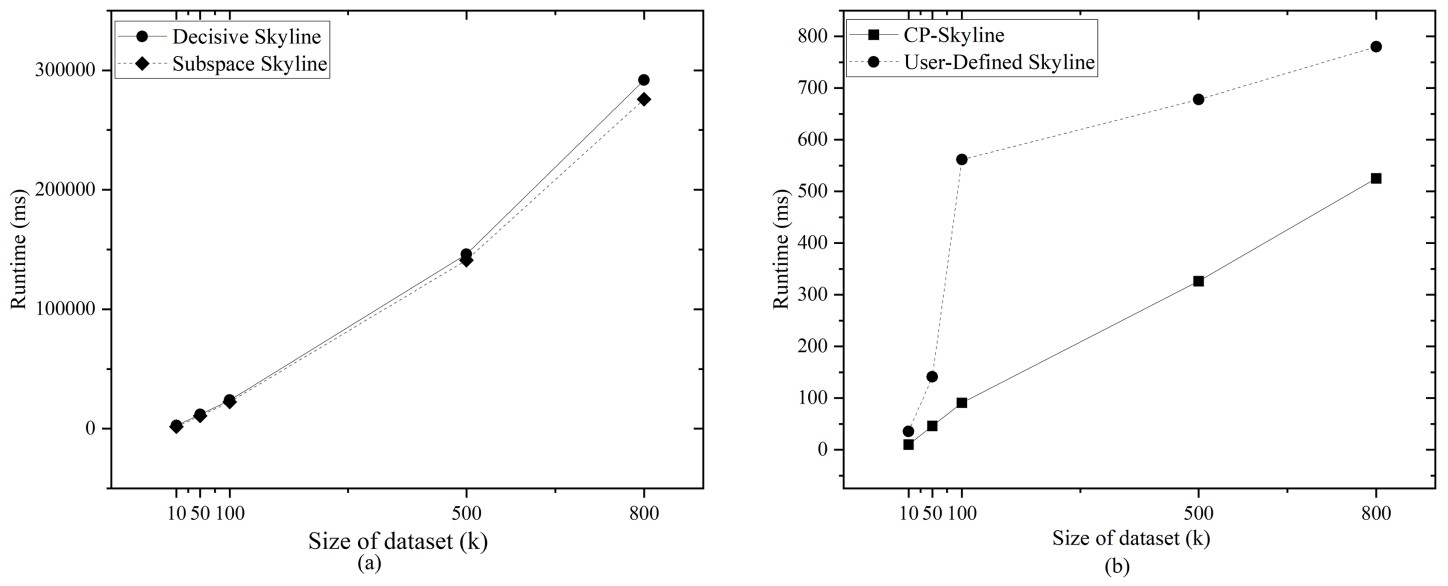

**Figure 10  (A, B) Query time *vs*. size of dataset (k) on Synthetic anti-correlated dataset.**

QWS (Fig. 9A) and NBA players (Fig. 9B). Finally, we analyze the changes of several query times with the size of the data set in the synthetic anti-correlation data set(Fig. 10).

Figures 7A and 7B shows the query time of different types of skyline query methods on QWS and NBA players, where in this case, the entire dataset was used as the candidate set. Traditional skyline queries were conducted with a full-dimensional query space, while

personalized skyline queries were performed with a five-dimensional query space. The traditional skyline query does not have any preference, so it needs to calculate the dominance relationship of all candidate data, so the query time is much higher than other personalized skyline query methods. Next, since the query time of subspace skyline and decisive skyline is much longer than that of CP-Skyline and user-defined skyline, Fig. 8A only describes the query time of decisive skyline and subspace skyline with a five-dimensional query space, Fig. 8B describes the query time of CP-Skyline and user-defined skyline with a five-dimensional query space. As the dataset size changes, we can see that CP-Skyline not only provides more detailed user's preferences than user-defined, but also spends less query time.

Figures 9A and 9B show the impact of the query dimension (attribute) involved in the skyline on the query time is examined on QWS and NBA players; it can be found that the query time of subspace skyline and decisive skyline is not much different, and the query time increases with the increase of query dimension. The reason is that the decisive skyline is based on the result of subspace skyline. The user-defined skyline and CP-Skyline reduce the query time with the increase of the dimension. The reason for this result is that with the increase of the query dimension, when the candidate data is pruned by the user's preference, a large number of data that the user is not interested in can be eliminated, which reduces the time spent on the skyline calculation, because the time complexity of the pruning is $O(n \cdot log(n))$, and the time complexity of the skyline query is $O(n^2)$. Due to the small amount of QWS data, less time is spent in the Skyline calculation phase at low latitudes, while the user-defined skyline and CP-Skyline require additional processing of user preferences, and then pruning candidate data based on user preferences. Pruning takes a relatively large proportion of time, and as the dimension increases, the time spent on pruning gradually decreases, and the advantages of using preference for pruning are gradually revealed, so there is a situation as shown in Fig. 9A.

Finally, we compared the query times of several methods as the data increases on the synthetic anti-correlated data set. Given that the query times for subspace skyline and decisive skyline are significantly longer than those for CP-Skyline and user-defined skyline, Fig. 10A focuses on the query times for decisive skyline and subspace skyline within a 5-dimensional query space. Meanwhile, Fig. 10B illustrates the query times for CP-Skyline and user-defined skyline within the same 5-dimensional query space. Similarly, because the advantage of CP-Skyline lies in its excellent pruning ability, CP-Skyline still has better performance. In general, a substantial portion of the computational time in skyline queries is allocated to the dominance calculation of the skyline. In personalized skyline queries, after incorporating user preferences, we can prune many tuples that do not meet the user's preferences. These tuples do not need to be included in the dominance calculation, thus reducing query time. Therefore, the more specific the user's preferences, the more tuples are removed during the pruning stage, and the less time it takes to obtain the Skyline set.

### Precision

Precision refers to the proportion of the number of documents that match the required information to the total number of documents retrieved. The precision indicates how

many data points in the skyline results returned by the query do meet the query conditions. High precision means that the returned results are credible and can be safely used for decision-making and analysis. High precision query results help users make more accurate decisions. Especially in important decision-making scenarios, achieving high precision in personalized skyline queries is critical for providing tailored and effective solutions. Since the all personalized skyline meets the user's preferences, the precision of the personalized skyline is 1. The difference is that CP-Skyline provides more refined preferences.

## CONCLUSION

Traditional personalized skyline query methods exhibit constraints in handling user's dependency preferences. Our work proposes the utilization of CP-Nets to address user's conditional preferences in the personalized skyline query process and define CP-Skyline. The CP-Skyline approach effectively incorporates user-defined conditional preference models, thereby narrowing down the query space during query processing. By establishing new dominance relationships for CP-Skyline computation, we can more accurately identify optimal solutions. Extensive experiments conducted on synthetic and real datasets distinctly confirm the significant advantages of the CP-Skyline method in enhancing skyline quality.

This research contributes a pragmatic and robust solution to the complexity of user's preferences in personalized decision support. However, ensuring feasibility and efficiency in practical applications remains a critical area for continued investigation. Additionally, The effectiveness of this method is contingent upon accurately defined preference information as specified by users, if the user's preferences are unclear or change frequently, it may affect the efficiency and accuracy of the algorithm. Further research should rigorously investigate methods for accurately deriving conditional user preferences from data and focus on protecting user's preference privacy in skyline queries.

### Funding

This research was supported by the National Natural Science Foundation of China (Grant No. 62362043, 62262036, 61962030), the Xingdian Talent Support Project (Grant No. KKXY202203008), and the Science and Technology Plan Projects of Yunnan Province (Grant No. 202205AF150003, 202204BQ040010). The funders had no role in study design, data collection and analysis, decision to publish, or preparation of the manuscript.

### Grant Disclosures

The following grant information was disclosed by the authors:
National Natural Science Foundation of China: 62362043, 62262036 and 61962030.
Xingdian Talent Support Project: KKXY202203008.
Science and Technology Plan Projects of Yunnan Province: 202205AF150003, 202204BQ040010.

## Competing Interests

The authors declare that they have no competing interests.

## Author Contributions

- Senfu Ke conceived and designed the experiments, performed the experiments, analyzed the data, performed the computation work, prepared figures and/or tables, authored or reviewed drafts of the article, and approved the final draft.
- Xiaodong Fu conceived and designed the experiments, performed the experiments, analyzed the data, prepared figures and/or tables, authored or reviewed drafts of the article, and approved the final draft.
- Jie Li conceived and designed the experiments, authored or reviewed drafts of the article, and approved the final draft.

## Data Availability

The raw measurements are available in the Supplemental Files.

## Supplemental Information

Supplemental information for this article can be found online at http://dx.doi.org/10.7717/peerj-cs.2659#supplemental-information.

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
