# Peer review of "Efficient skyline query processing with user-specified conditional preference"

_PeerJ Computer Science, doi:10.7717/peerj-cs.2659_

## Round 0.1 · original submission · Major Revisions

With respect to the reviewers’ comments and my reading of the paper, the paper presents a good idea and contributes to the field of personalized skyline query processing. However, the reviewers have raised several issues concerning a lack of clarity and discussion in the details of the experimental design. I have therefore decided that the manuscript requires major revisions before it could be reconsidered for publication in the journal. Please respond, point by point, to the issues raised by the reviewers, making clear any changes in the manuscript.

Reviewer 1 ·

Basic reporting

The idea presented in this article is very interesting, and the author has also written it in clear and good English. The article has effectively presented the introduction and background to demonstrate how the work fits into the broader field of knowledge.
However, this research has not yet provided a detailed explanation of the extent to which "not oversimplifying user preferences" can be accommodated. It would be better if the authors could provide a formal definition of this concept.

The data structure used for the implementation of CPT has not been detailed. A formal definition regarding this should be added.

An illustration should be provided if there are multiple user preferences, to clearly show how CP-nets and CPT are created for each user's preferences.

In the sentence on line 47, why is h5 not included in the range?

The statements on lines 233 and 234, in my opinion, require more detailed explanation.

The mechanism for creating the induced graph needs to be clarified, including what data structure is used for its implementation, how the induced graph is formed, and what the arrows in the induced graph represent.

I suggest creating an illustrative case that can be followed from start to finish, providing examples for different user preferences, showing how each user can obtain their desired skyline.

The statements on lines 241 and 242 are somewhat confusing; please provide a more detailed explanation.

Experimental design

How is precision calculated in section 5.2.3, and how can the authors claim the high precision obtained?

Validity of the findings

Based on the experiments conducted, could the authors provide insights on when this algorithm can achieve its best performance? It would be better if the experiments also considered the complexity of the preferences included, so that the audience can take into account the complexity of the preferences when using this algorithm.

Cite this review as

·

Basic reporting

This study investigates the skyline query processing with an interesting property of user specific conditional preferences Conditional Preference Networks (CP-Nets). The proposed CP-Skyline method seeks to enhance the efficiency and effectiveness of skyline queries, particularly in scenarios where user preferences are complex and involve interdependencies between attributes. Through extensive experiments on synthetic and real-world data, the authors show that using their algorithm is a better way of improving skyline quality.

The manuscript is well organized and technically sound, proposing a new solution in processing personalized skyline queries. The writing is mainly clear and focused, but some sections may need greater clarity of language. This paper makes a valuable contribution in terms of formalizing the problem and devising an efficient algorithm, but had scope for improvement when it comes to explaining computational complexity as well demonstrating experimental results more extensively. Secondly, the manuscript requires some minor typographical touch ups and formatting adjustments to reflect true publication status.

Some detailed comments:
a. Example 1 could be more integrated into the main text rather than being presented separately. Embedding it within the another intensive discussion can provide a better narrative,

b. The paper's contributions should be more fluidly incorporated into the problem-solving narrative. Instead of listing them, consider embedding these contributions within the discussion of how the paper addresses specific challenges in skyline query processing, thereby enhancing readability and emphasizing the novelty of the approach.

c. The introduction should reference the most closely related work on user preference-based skyline queries. Including key studies (I think there exists some user-preferences-based skyline paper) would better position this paper within the existing literature and highlight its contributions to the field.

e. The problem statement on line 248 should be formally defined as Definition 5 to emphasize its significance. This formal presentation aligns with academic conventions and makes the concept more accessible for readers who may need to refer back to it later in the paper.

f. The writing style could be more formal and precise. Avoid contractions and overly complex sentences to ensure clarity. Consider a thorough revision to enhance readability, possibly with the assistance of a professional editor.

g. Several typographical errors and stylistic inconsistencies were noted throughout the manuscript.

- On line 266, it should read "stage 1 involves ..." instead of the current phrasing.
- On line 297, the phrase "is returned(Represented" is unclear and should be clarified; does it mean that the value is returned and represented by a specific symbol? The meaning needs to be made explicit.
- Contractions such as "don't" should be replaced with "do not" for consistency with the formal tone of the paper.
- On line 336, where "G(Line 5)" appears without proper explanation—ensure that such references are clear and contextually appropriate.
- In line 341, the term "Hierarchies" should be italicized to indicate that it is a parameter.

Experimental design

a. The computational complexity analysis should be expanded with more detailed explanations. Instead of briefly mentioning big-Oh notation, the authors may provide a step-by-step breakdown of their derivation, linking them to specific parts of the algorithm to improve understanding.

b. The experimental results section (lines 375-381) should specify the number of rows and columns for each dataset. Presenting this information in a better format would improve clarity and allow readers to better understand the performance implications of the dataset.

c. Graphical representations should use distinct symbols and line types for different elements to improve clarity. For instance, using solid and dashed lines for different algorithms within the same graph can help differentiate them, making the results easier to interpret by the reader.

d. I've found some anomalies in Figures 6b and 9a. Consider to put more deeper analysis. Exploring possible causes, such as data characteristics or experimental setup limitations, would enhance the credibility of the findings and demonstrate a thorough understanding of the results.

e. Provide details on memory or RAM usage for each environmental setup, particularly the feasibility of processing large datasets with 8GB of RAM. This information would offer valuable insights into the practical scalability of the proposed method.

f. Have you tried to deal with Anti-Correlated synthetic dataset? Please provide it. This data is hard to process and there are number of previous skyline studies that find a difficulty in both runtime and resource aspects to handle this kind of dataset.

Validity of the findings

While the idea presented in this paper is quite good and makes a contribution to the field of personalized skyline query processing, the manuscript has significant room for improvement One of my questions is whether there any possibility for your hierarchy-based dominance to form a cyclic dominance relation? If not, that's good. Please justify that thing.

Also, the issues raised in the detailed comments suggest that a major revision is necessary to enhance the quality of the paper. Please also improve the experimental result part as suggested above. The authors should carefully consider revising the manuscript to address these points to ensure that it meets the high standards expected for publication in a top journal.

External reviews were received for this submission. These reviews were used by the Editor when they made their decision, and can be downloaded below.

---

## Round 0.2 · Minor Revisions

The reviewers find the paper much improved. However there are still some minor issues. As to the validity of the findings, please clarify the impact of adding preferences to the algorithm, its application to real world scenarios, acyclic nature and limitations. Please also address minor concerns relating to the manuscript and academic language.

Reviewer 1 ·

Basic reporting

This article has undergone significant improvements, particularly in the aspects of definitions and detailed explanations regarding the data structures and the examples used to clarify the proposed algorithm. Overall, I believe the manuscript now meets the necessary criteria for publication. The English used is clear, unambiguous, and professional throughout. Relevant literature references have also been adequately included, providing sufficient context in the related field.

Moreover, the article is well-structured, utilizing tables and figures that aid the reader's understanding. Raw data has also been shared, enhancing the transparency of the research findings. The article stands on its own with relevant results that support the proposed hypotheses. The formal results include clear definitions of all terms and theorems, with proofs elaborated in detail to ensure accuracy and clarity. Thank you for accommodating the feedback I provided in the previous review.

Experimental design

In the methods and experiment section, the author has provided detailed explanations of the steps used and the experimental scenarios conducted. The research question is well-defined, relevant, and meaningful. The author has also explained how the research fills the identified knowledge gap.

The experiments were well-conducted, and each result is explained in detail. The methods are described with sufficient detail and information, making them replicable.

Validity of the findings

All underlying data has been provided; the data is robust, statistically valid, controlled, and explained in detail. In the experiment section, I suggest that the author clarify how the addition of preferences to this algorithm impacts the time and number of skylines produced. Does this addition affect time and the number of skylines linearly, or otherwise? Besides that, I believe what the author has presented sufficiently demonstrates the reliability of the algorithm. The conclusions are also well stated and connected to the research question.

Additional comments

This paper has been revised very well, making it easy for readers to follow the explanations within. The algorithm is also accompanied by its implementation on GitHub, which makes it easier for readers to explore and learn. This algorithm will be highly beneficial for skyline query research, particularly in the area of personalized skyline. To further enhance the development of this algorithm, I recommend that the author include explanations of how the algorithm can be applied in various real-world scenarios. This would improve the practical understanding and relevance of the proposed method.

Cite this review as

·

Basic reporting

The authors have made substantial improvements to the basic reporting aspects:

- Example 1 has been successfully integrated into the main text, enhancing readability
- The paper's contributions are now naturally woven into the problem-solving narrative
- The related work section has been expanded to better contextualize user-preferences-based skyline queries
- Definition 5 has been formally added as suggested
- Most typographical and stylistic issues have been addressed

Some remaining minor concerns:

A few sections still contain informal language that could be made more academic
Some technical terms would benefit from more precise definitions

Experimental design

The experimental section shows important improvement:

- Detailed dataset information has been added, including row and column specifications
- Graphical representations now use distinct line types for better clarity
- The anomalies in Figures 6b and 9a have been explained in detail
- Anti-correlated synthetic dataset experiments have been included
- Computational environment specifications have been clarified (Intel® Xeon® Gold 6330, 80G RAM)

The computational complexity analysis, while expanded, could still benefit from clear discussion of worst-case scenarios and their likelihood in real applications

Validity of the findings

The acyclic nature explanation helps, though I think readers would benefit from a quick example showing why cycles can't happen. It's nice to see the method works well on anti-correlated data - that's often a stumbling block.
A brief note about any limitations would be helpful - every method has them, and being upfront about them helps readers make informed choices.

Additional comments

For language and Presentation:
Final polishing of academic language is recommended
Some technical terms could use more precise definitions


Recommendations for minor revision:
Strengthen theoretical foundations of complexity analysis
Provide clearer examples of the acyclic nature
Polish academic language
Add brief discussion of method limitations

---

## Round 0.3 · accepted · Accept

Reviewer 1 is happy that the authors have addressed all their concerns. I have assessed the revision myself and am happy with the current version and consider the manuscript ready for publication.

Reviewer 1 ·

Basic reporting

The manuscript has undergone significant improvements, particularly in terms of definitions and detailed explanations regarding the data structures and examples used to clarify the proposed algorithm. Based on my previous review, the article already meets the criteria for publication. The English is clear, unambiguous, and professional throughout. Relevant literature references are adequately included, providing sufficient context in the related field.

The manuscript is well-structured, with tables and figures effectively supporting the reader's understanding. Raw data has been shared, ensuring transparency in the research findings. The article is self-contained, with relevant results that support the proposed hypotheses. All terms and theorems are clearly defined, and the proofs are detailed and accurate.

Experimental design

The methods section is detailed, and the experimental scenarios are well-explained. The research question is clearly defined, relevant, and meaningful. The manuscript addresses a significant knowledge gap in the field.

The experiments were conducted rigorously, and the results are explained clearly. The methods are described in sufficient detail to allow replication.

Validity of the findings

The data provided is robust, statistically valid, and well-explained. In my previous review, I suggested clarifying the impact of preferences on the time and number of skylines produced by the algorithm. The revised manuscript adequately addresses this point, and I believe the findings demonstrate the reliability and validity of the algorithm. The conclusions are well-stated and aligned with the research question.

Additional comments

The revisions made to the manuscript have significantly improved its quality, making it easier for readers to follow the explanations. The accompanying implementation on GitHub enhances accessibility and usability for readers interested in exploring the algorithm further.

The manuscript is now well-prepared for publication. From my perspective, the current version addresses all relevant points, and no further revisions are needed. Thank you for accommodating the feedback provided in the earlier review.

Cite this review as